# Pan-Canadian Analysis of Practice Patterns in Small Cell Carcinoma of the Cervix: Insights from a Multidisciplinary Survey

Kevin Yijun Fan [1,2], Rania Chehade [1,2], Andrew Yuanbo Wang [3], Anjali Sachdeva [1], Helen J. MacKay [1,2,†] and Amandeep S. Taggar [1,2,*,†] 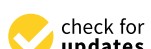

1  Temerty Faculty of Medicine, University of Toronto, Medical Sciences Building, 1 King's College Cir., Toronto, ON M5S 1A8, Canada; kevin.fan3@uhn.ca (K.Y.F.); rania.chehade@uhn.ca (R.C.); anjali.sachdeva@mail.utoronto.ca (A.S.); helen.mackay@sunnybrook.ca (H.J.M.)
2  Department of Radiation Oncology, Sunnybrook Odette Cancer Centre, T-wing 2075 Bayview Avenue TG 260, Toronto, ON M4N 3M5, Canada
3  Schulich School of Medicine and Dentistry, University of Western Ontario, 1151 Richmond St., London, ON N6A 5C1, Canada; awang2025@meds.uwo.ca
*  Correspondence: aman.taggar@sunnybrook.ca; Tel.: +1-416-275-9108
†  These authors contributed equally to this work.

**Abstract:** Small-cell neuroendocrine carcinoma of the cervix (SCNECC) is a rare cancer with poor prognosis, with limited data to guide its treatment. The objective of this study was to evaluate practice patterns in the management of SCNECC. A 23-question online survey on management of SCNECC was disseminated to Canadian gynecologic oncologists (GO), radiation oncologists (RO) and medical oncologists (MO). In total, 34 practitioners from eight provinces responded, including 17 GO, 13 RO and four MO. During staging and diagnosis, 74% of respondents used a trimodality imaging approach, and 85% tested for neuroendocrine markers. In early-stage (1A1-1B2) SCNECC, 87% of practitioners used a surgical-based approach with various adjuvant and neoadjuvant treatments. In locally advanced (1B3-IVA) SCNECC, 53% favored primary chemoradiation, with cisplatin and etoposide, with the remainder using surgical or radiation-based approaches. In metastatic and recurrent SCNECC, the most common first-line regimen was etoposide and platinum, and 63% of practitioners considered clinical trials in the first line setting or beyond. This survey highlights diverse practice patterns in the treatment of SCNECC. Interdisciplinary input is crucial to individualizing multimodality treatment, and there is a need for prospective trials and intergroup collaboration to define the optimal approach towards managing this rare cancer type.

**Keywords:** cervical; neuroendocrine; chemotherapy; radiation; small-cell carcinoma of the cervix; neuroendocrine carcinoma of the cervix

## 1. Introduction

Neuroendocrine tumors (NETs) of the female genital tract are a rare entity, with the cervix being the most common site. Based on the 2014 WHO classification, cervical NETs are classified into low-grade NET, adenocarcinoma admixed with neuroendocrine carcinoma, and high-grade neuroendocrine carcinoma of the cervix (HGNECC) [1]. HGNECC makes up 1–1.5% of all cervical cancers and is composed of small-cell neuroendocrine carcinoma of the cervix (SCNECC, approximately 80% of all HGNECC) and large-cell neuroendocrine carcinoma of the cervix (LCNECC) [1]. Most clinical guidelines and expert consensus statements [2–5] combine recommendations regarding the management of both SCNECC and LCNECC, without making a clear distinction between the two.

Although SCNECC is ultimately a morphological diagnosis, the expression of neuroendocrine markers including synaptophysin, chromogranin, CD56 and neuron-specific enolase (NSE) often supports the diagnosis [6], and most cases are human papilloma virus

(HPV)-associated [7]. SCNECC has an aggressive biology and natural history, and is associated with early nodal involvement and metastatic spread, high rates of local and distant relapse, and poor prognosis, with a 5-year overall survival (OS) of 34% [8].

Due to its rarity, there is a lack of high-quality data and prospective trials to guide its management, and most of its treatment approaches are extrapolated from the treatment of cervical or small-cell lung cancer. In order to reach a consensus regarding the optimal management of this rare disease, there is a need to understand real-world practices, with the aim of identifying gaps in clinical evidence and knowledge translation. The objective of this study was to gather and analyze the practice patterns of Canadian gynecologic oncologists (GO), radiation oncologists (RO) and medical oncologists (MO) in the treatment and management of SCNECC, through a national survey. Furthermore, we aimed to compare these practices to available established guidelines, and to form a basis for a consensus statement regarding the diagnosis and treatment of SCNECC.

## 2. Materials and Methods

A search on Medline and Embase was conducted for articles pertaining to the treatment of SCNECC. The literature-based approach was applied to identify key themes in the treatment of SCNECC, and subsequently a 23-question online survey (Supplementary Materials S4) regarding practice patterns in the management and treatment of SCNECC was developed using SurveyMonkey™ (San Mateo, CA, USA). Survey questions were written in English and followed the format of either simple multiple-choice or multiple-choice matrix. All questions regarding treatment options had the answer option of "I don't know" as well as the opportunity to enter an alternative treatment option in free-text format.

Between 29 August 2023 and 28 September 2023, this survey was disseminated via email among gynecological oncologists and medical oncologists in the Society of Gynecologic Oncology of Canada (GOC), as well as a database of Canadian radiation oncologists who specialize in the treatment of gynecological malignancies. All respondents were actively practicing physicians in Canada who indicated that they had experience in treating SCNECC. Participation was voluntary and non-incentivized, and all responses were anonymized. All participants provided written consent for their anonymous responses to be used for research purposes. Categorical variables were compared using Chi-squared test with an alpha ($\alpha$) of 0.05. This project received a score of 2 (minimal risk) based on ARECCI Ethics Screening tool [9], which is a Canadian questionnaire-based risk assessment score that determines the level of risk, types of ethical risks, and the appropriate type of ethics review for research and quality improvement projects.

## 3. Results

Thirty-four practitioners responded to this survey. Practitioners represented eight Canadian provinces: Ontario (n = 8, 24%), Quebec (n = 6, 18%), British Columbia (n = 6, 18%), Central Canadian provinces (n = 10, 30%), and Atlantic Canadian provinces (n = 4, 12%). Among practitioners who chose to respond to the optional question "Please enter the name of your primary center of practice", 17 centers were represented. Most practitioners (n = 30, 88%) indicated that between one and four cases of SCNECC were treated at their institution annually. Practitioners responding to this survey included 17 gynecological oncologists (50%), 13 radiation oncologists (38%) and four medical oncologists (12%). They had varying levels of practice experience, with 22 (64%) having more than ten years of practice experience (Table 1). In total, 30 (88%) respondents self-identified as practicing in an academic institution (15/17 gynecologic oncologists, 12/13 radiation oncologists and 3/4 medical oncologists). Of the 34, 32 (94%) respondents indicated that all three specialties were available where they practice; 56% (19/34) of respondents indicated that systemic therapy for SCNECC is prescribed by gynecologic oncologists at their center, whereas 44% (15/34) of respondents indicated that it is prescribed by medical oncologists.

**Table 1.** Baseline characteristics of institutions and practitioners treating small-cell neuroendocrine carcinoma of the cervix (SCNECC) in Canada (n = 34 practitioners across 8 provinces).

| Institution Characteristics | | N | % |
|---|---|---|---|
| | Ontario | 8 | 24 |
| | Quebec | 6 | 18 |
| | British Columbia | 6 | 18 |
| | Alberta | 5 | 15 |
| Province | Manitoba | 3 | 9 |
| | Newfoundland | 2 | 6 |
| | Nova Scotia | 2 | 6 |
| | Saskatchewan | 2 | 6 |
| Academic institution | Yes | 30 | 88 |
| | No | 4 | 12 |
| | Gynecologic oncology | 32 | 94 |
| Specialties available | Radiation oncology | 34 | 100 |
| | Medical oncology | 33 | 97 |
| Primary specialty that delivers systemic therapy | Gynecologic oncology | 19 | 56 |
| | Medical oncology | 15 | 44 |
| | Radiation oncology | 0 | 0 |
| | 0 | 1 | 3 |
| Number of patients with SCNECC treated annually | 1 to 4 | 30 | 88 |
| | 5 to 10 | 2 | 6 |
| | >10 | 1 | 3 |
| Practitioner Characteristics | | N | % |
| | Gynecologic oncology | 17 | 50 |
| Primary specialty of practice | Medical oncology | 4 | 12 |
| | Radiation oncology | 13 | 38 |
| | ≤5 | 6 | 18 |
| Number of years in practice | 6 to 10 | 6 | 18 |
| | 11 to 15 | 11 | 32 |
| | ≥16 | 11 | 32 |

In the initial staging of newly diagnosed SCNECC, most practitioners took a multi-modality approach, with 97% (n = 33) utilizing MRI of the pelvis, 88% (n = 30) utilizing 18F-FDG-PET/CT (positron emission tomography), 82% (n = 28) utilizing CT of the chest, abdomen and pelvis, and 74% (n = 25) utilizing all three of the above modalities. Only one practitioner (3%) utilized EUS (endoscopic ultrasound) with sigmoidoscopy and cystoscopy (Figure 1A). Most practitioners used a selective approach to decide whether to include brain imaging in their initial diagnostic workup, with 44% (n = 15) obtaining brain imaging based on the symptoms or presence of lung metastases, 26% (n = 9) doing so based on symptoms alone, and 9% (n = 3) doing so based on the presence of lung metastases alone, whereas 21% (n = 7) routinely obtained brain imaging for all patients (Figure S1).

The most commonly obtained immunohistochemical markers at the time of diagnosis were p16 (94%, n = 32), synaptophysin (79%, n = 27), p53 (79%, n = 27), chromogranin (76%, n = 26), and HPV (62%, n = 21), whereas mismatch repair (MMR) (50%, n = 17), programmed death-ligand 1 (PD-L1) (44%, n = 15), CD56 (24%, n = 8), and NSE (18%, n = 6) were obtained less frequently. Only three respondents (9%) routinely obtained next-generation sequencing (NGS) panels at the time of initial diagnosis (Figure 1B).

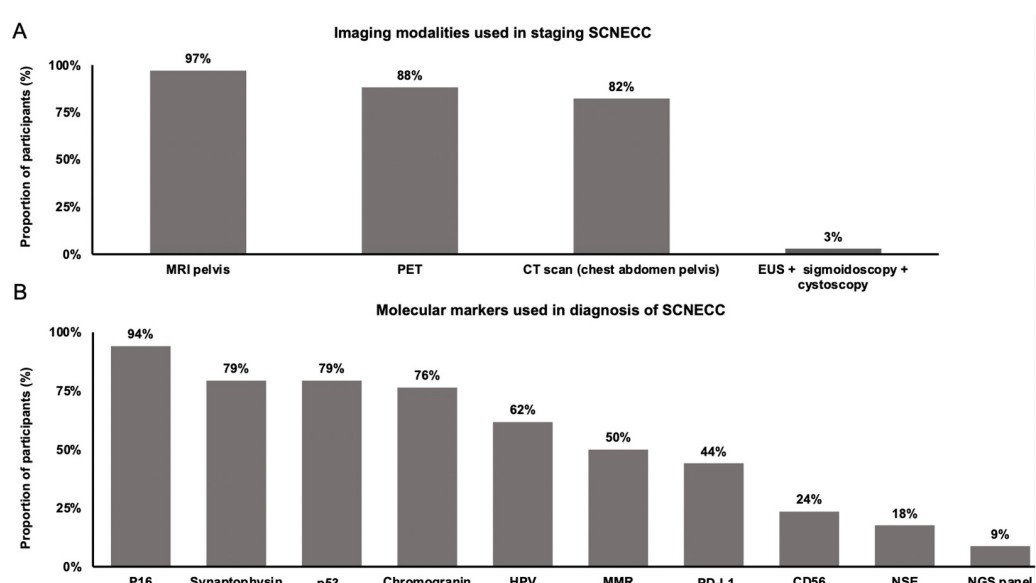

**Figure 1.** Practice patterns in staging and diagnosis of SCNECC. (**A**) Imaging modalities used in the staging of SCNECC. Abbreviations: MRI (magnetic resonance imaging), PET (18F-FDG-PET/CT; positron emission tomography), EUS (endoscopic ultrasound). (**B**) Molecular markers used in the diagnosis of SCNECC. Abbreviations: P16 (cyclin-dependent kinase inhibitor 2A), p53 (tumor protein p53), HPV (human papilloma virus), MMR (mismatch repair), PD-L1 (programmed death-ligand 1), NSE (neuron specific enolase), NGS (next-generation sequencing).

In early-stage SCNECC (FIGO 2018 stage IA1-1B2), the majority of practitioners (n = 26, 87%) used a surgical approach. Among practitioners who used a surgical approach, 14 (54%) used adjuvant chemotherapy, 10 (38%) used neoadjuvant chemotherapy, 6 (23%) used adjuvant radiotherapy, 2 (8%) used adjuvant chemoradiotherapy, and 1 (4%) used neoadjuvant chemoradiotherapy. A non-surgical approach (i.e., concurrent chemoradiotherapy plus brachytherapy, with or without additional chemotherapy) was utilized by a smaller subset of practitioners (n = 4, 13%) (Figure 2A).

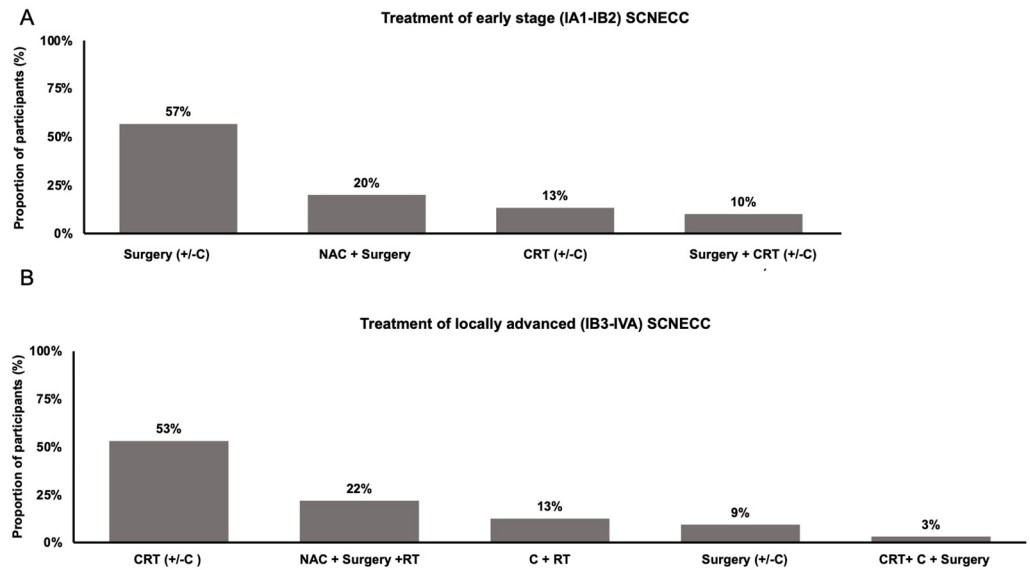

**Figure 2.** Practice patterns in the treatment of (**A**) early-stage and (**B**) locally advanced SCNECC. Abbreviations: C (chemotherapy), +/− (with or without), NAC (neoadjuvant chemotherapy), + (followed by), CRT (concurrent chemoradiation with external beam radiation +/− brachytherapy), RT (radiotherapy).

In locally advanced (FIGO 2018 stage 1B3-IVA) SCNECC, there was more variability in practice. The majority of practitioners (n = 17, 53%) utilized a concurrent chemoradiation and brachytherapy-based approach, among which 13 (76%) prescribed additional chemotherapy. Eleven practitioners (34%) utilized a surgical approach, among whom nine (82%) used neoadjuvant chemotherapy, seven (64%) used adjuvant radiation, two (18%) used adjuvant chemotherapy, and one (9%) used neoadjuvant chemoradiation. Four practitioners (13%) utilized chemotherapy followed by radiotherapy (Figure 2B).

When utilizing curative intent concurrent chemoradiation in the treatment of non-metastatic SCNECC, most practitioners (n = 22, 69%) prescribed etoposide and platinum (EP), whereas ten practitioners (31%) prescribed single-agent platinum (Figure 3A) concurrently with radiation. Most practitioners (n = 27, 84%) prescribed additional chemotherapy after concurrent chemoradiation, with the most common regimen (n = 25, 93%) being EP (Figure 3B), and the majority (n = 24, 89%) prescribed between four and six cycles of additional chemotherapy (Figure 3C).

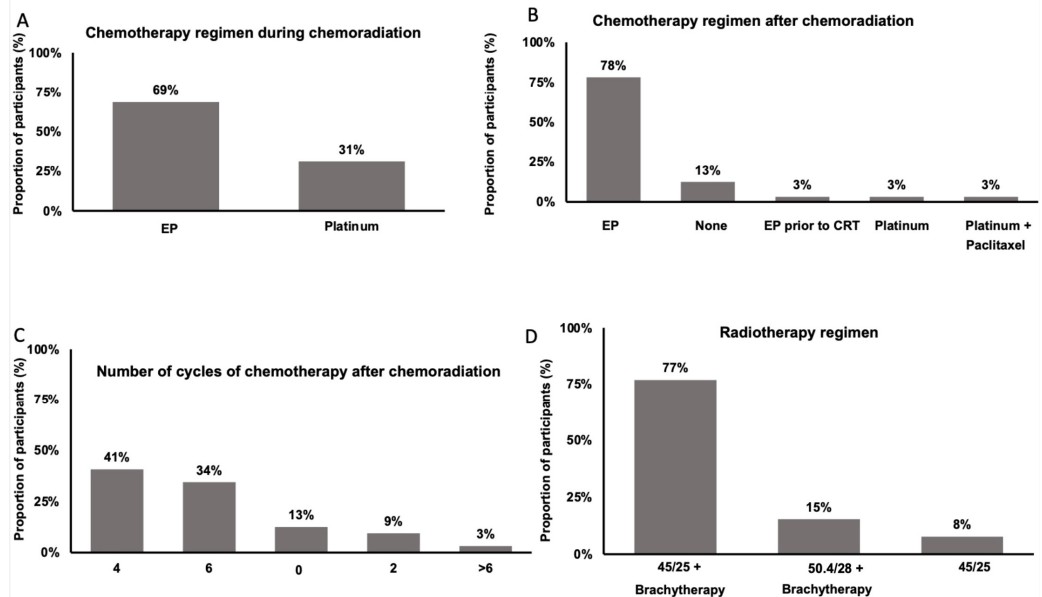

**Figure 3.** Chemotherapy and radiotherapy regimens used in the definitive treatment of SCNECC. (**A**) Chemotherapy regimen used during definitive chemoradiation, (**B**) chemotherapy regimen used after definitive chemoradiation, (**C**) number of cycles of chemotherapy used after definitive chemoradiation, and (**D**) dose-fractionation regimen used in definitive radiotherapy. Abbreviations: EP: etoposide and platinum. CRT: concurrent chemoradiation. 45/25: 45 Gray in 25 fractions. 50.4/28: 50.4 Gray in 28 fractions.

When utilizing radiotherapy for the curative treatment of SCNECC, the most common dose fractionation regimen was 45 Gray in 25 fractions with brachytherapy (n = 20, 77%), and other regimens included 50.4 Gray in 28 fractions with brachytherapy (n = 4, 15%), and 45 Gray in 25 fractions without brachytherapy (n = 2, 8%) (Figure 3D). Brachytherapy was delivered based on clinical presentation as per institutional practices.

In newly diagnosed metastatic (FIGO 2018 stage IVB) SCNECC, in the first line setting, the most commonly employed treatment was EP chemotherapy (n = 21, 81%). Other regimens included platinum, paclitaxel, bevacizumab and pembrolizumab (n= 2, 8%), and paclitaxel, topotecan and bevacizumab (n = 1, 4%). In the second- or third-line setting, the most used regimens were platinum and paclitaxel (n = 10, 25%), platinum, paclitaxel, topotecan and bevacizumab (n = 4, 10%) and platinum, paclitaxel, bevacizumab and pembrolizumab (n = 7, 18%). Other combinations included re-treatment with EP (n = 3, 8%), immunotherapy (n = 3, 8%), and vincristine, doxorubicin and cyclophosphamide (n = 2, 5%). Eight practitioners (20%) suggested enrolment in clinical trials in the second- or

third-line setting, and three (8%) suggested best supportive care with no further systemic therapy (Figure 4A).

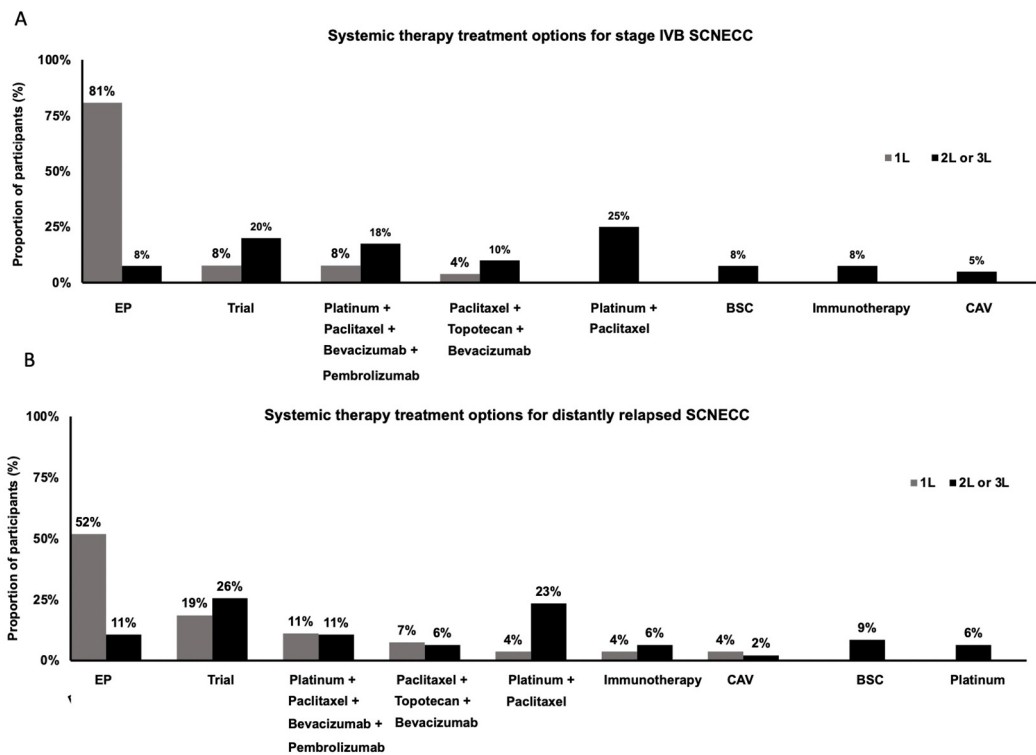

**Figure 4.** Practice patterns in systemic therapy treatment options of metastatic (**A**) and distantly relapsed (**B**) SCNECC. Abbreviations: EP (etoposide and platinum), BSC (best supportive care), CAV (cyclophosphamide, doxorubicin, vincristine), 1L (first line), 2L (second line), 3L (third line).

In distantly recurrent SCNECC, the four most common first-line treatment options were the same as those used in de novo metastatic SCNECC. Additional first-line options included platinum and paclitaxel (n = 1, 4%), immunotherapy (n = 1, 4%), and vincristine, doxorubicin and cyclophosphamide (n = 1, 4%) (Figure 4B). Two practitioners (8%) commented that the choice of therapy would depend on the time to relapse. In total, 37% (10/27) of practitioners indicated that they would enroll a patient with metastatic SCNECC in a clinical trial in either the first-, second- or third-line setting, and 63% of practitioners (17/27) indicated that they would enroll a patient with distantly recurrent SCNECC in a clinical trial in either the first-, second- or third-line setting (Figure 4A,B).

When treating a patient with local relapse after curative intent concurrent chemoradiation, the most common approaches used were surgery and chemotherapy (n = 12, 41%), followed by chemotherapy alone (n = 7, 24%), surgery alone (n = 7, 24%), and radiation and chemotherapy (n = 3, 10%) (Figure S2A). When treating a patient with regional nodal relapse after curative intent concurrent chemoradiation, the most common approach used was chemotherapy alone (n = 21, 68%), followed by radiation and chemotherapy (n = 8, 24%), surgery (n = 1, 3%), or surgery plus chemotherapy (n = 1, 3%) (Figure S2B).

After curative intent treatment of nonmetastatic SCNECC, most practitioners followed a surveillance regimen every 3 to 6 months with various imaging modalities: CT (n = 12, 35%), 18F-FDG-PET/CT (n = 8, 24%), MRI (n = 1, 3%), or history and physical alone without routine imaging (n = 12, 35%), whereas one practitioner (3%) followed a yearly surveillance regimen with CT (Figure S3).

## 4. Discussion

In this survey of Canadian physicians, we identified variable practice patterns in the management of SCNECC. In early-stage disease, most practitioners used a surgical

approach with various adjuvant and neoadjuvant treatments. In locally advanced disease, many practitioners used a chemoradiation or radiation-based approach, although other practitioners also employed a surgical approach. In the recurrent and metastatic setting, the most common first-line regimen was EP chemotherapy. Below we will compare participants' responses with recommendations from published clinical guidelines and evidence from published literature.

In the staging and diagnosis of SCNECC, most respondents in our survey used a tri-modality approach, which is in line with recommendations made by the Society of Gynecological Oncology (SGO) [2] and the NCCN [10]. Most respondents also selectively utilized brain imaging, which is in line with the SGO and MDACC (MD Anderson Cancer Centre), which both recommend selective brain imaging (i.e., for patients with distant metastases (brain or lung) or neurological symptoms [2,4]). The diagnosis of SCNECC can be made based on morphology alone, although immunohistochemical markers can often support the diagnosis [2,6]. Furthermore, 85% of cases are HPV-positive [7]. Indeed, most respondents in our survey tested for one or more neuroendocrine markers, as well as HPV. Although only 5% of SCNECC are positive for PD-L1 [11], in our survey, approximately half of respondents still tested for this marker; however, this was typically in the setting of guiding palliative systemic therapy rather than in initial diagnosis.

In early-stage SCNECC, most guidelines recommend surgery, with the SGO [2] and the NCCN [10] recommending either adjuvant chemotherapy or adjuvant chemoradiation, and the MDACC [4] recommending both adjuvant therapies in sequence. In concordance with these guidelines, most practitioners in our survey agreed on using a surgical approach. Following surgery, the majority of respondents in our survey prescribed adjuvant chemotherapy, and this has been shown to improve PFS [12] and OS [13]. Many respondents also prescribed adjuvant radiotherapy; however, a recent metanalysis [14] did not show an improvement in overall recurrence or survival with this approach.

In locally advanced SCNECC, the SGO [2], MDACC [4], and NCCN [10] all recommend chemoradiation with brachytherapy and additional chemotherapy. In our survey, only 53% of respondents used the recommended chemoradiotherapy-based approach, while 34% used a surgical approach. Indeed, retrospective studies have shown conflicting evidence when comparing radiotherapy-based and surgical approaches in both early and locally advanced SCNECC [15–18], and prospective evidence is needed to determine the optimal definitive treatment modality. When prescribing concurrent chemoradiation, respondents in our survey most often prescribed concurrent EP, followed by 4–6 cycles of additional EP. This approach is in line with guidelines [2,4], and has been associated with improved cancer-specific survival [19]. Finally, in line with the above guidelines, most respondents utilized brachytherapy in addition to external beam radiotherapy, although there are conflicting data as to whether this approach improves OS [18,20].

In metastatic and distantly recurrent SCNECC, most respondents in our survey prescribed EP, which is in line with most guidelines [2,4]. The addition of immunotherapy to EP has been extrapolated from the treatment of small-cell lung cancer [21,22]; however, no respondents selected this approach, possibly related to funding limitations in Canada. Topotecan, bevacizumab and paclitaxel were associated with an improvement in progression-free survival, but not OS, compared to other regimens in recurrent HGNECC [23], and only a small proportion of respondents in our survey utilized this option. In addition, a small proportion of respondents in our survey utilized single-agent immunotherapy, mostly in the second line setting or beyond, and a small prospective study found a median PFS of only 2.1 months with this approach [24]. Finally, genomic studies have demonstrated a high rate of actionable mutations in SCNECC, ranging from 50% to 73% [25–27]; however, only a small proportion of respondents in our survey performed NGS testing, again perhaps related to funding limitations in Canada. This indicates a need for increased access to precision medicine approaches and enrolment in clinical trials.

The strengths of this study include: The multidisciplinary input in the creation, response and analysis of this survey, the anonymous and voluntary nature of responses,

and the flexible format of individual questions (i.e., including option "other [please specify]" to allow for responses outside of the pre-specified multiple choices). Furthermore, the single-payer universal health insurance system makes Canada an ideal country to objectively study practice patterns without the confounding factor of financial biases in resource allocation.

A limitation of this study was that most participants only specialized in one treatment modality (i.e., surgery, radiation or chemotherapy), yet all participants were required to answer questions pertaining to all three modalities. This was in part reconciled by providing the answer choice "I don't know" for selected questions pertaining to systemic therapy options in the palliative setting, which may be esoteric to providers who primarily prescribe locoregional therapy. Responses from clinicians in each specialty are included in Supplementary File S5. In addition, as this was an anonymous survey, the institution name was not available for all respondents, hence the survey was analyzed based on the number of individual respondents rather than the number of institutions. As respondents were encouraged to respond on behalf of their institution, the number of respondents likely underrepresents the total number of clinicians that these results represent. Since this survey was disseminated throughout the Society of Gynecologic Oncology of Canada, it was not possible to record the total number of clinicians that had access to our survey, hence we were unable to report an exact response rate. Finally, the logistics and resources available in a publicly funded healthcare system need to be taken into consideration when generalizing these results.

There is currently no Canadian consensus guideline on the management of SCNECC, and this study represents the first national physician survey on this topic. In early-stage and locally advanced SCNECC, there needs to be further prospective evidence to guide the optimal choice and sequencing of treatment modalities in definitive treatment. Multidisciplinary discussion and shared decision making, for example through institutional and regional tumor boards, is essential to exchange and synthesize expert opinions in the management of this rare tumor type. In the metastatic and recurrent setting, the use of NGS-based precision medicine approaches and enrolment in clinical trials should be encouraged, and there is an urgent need for prospective trials. Overall, this study highlights the need for consensus guidelines informed by outcome data with intergroup collaboration, which are essential for rare tumors such as SCNECC.

## 5. Conclusions

In a multidisciplinary group of physicians surveyed across Canada, many practice patterns were in line with recommendations from existing consensus guidelines, although there was also variability in practice. In initial staging and diagnosis, most practitioners use a tri-modality imaging approach and test for neuroendocrine markers. In early-stage and locally advanced SCNECC, most practitioners agree on pursuing multi-modality treatment; however, there is variability in the selection and timing of treatment modalities. In metastatic and recurrent SCNECC, most practitioners use first-line EP, and many practitioners consider clinical trials in the first-, second- or third-line setting. SCNECC is a rare cancer type requiring multidisciplinary input and prospective evidence to guide optimal management.

**Supplementary Materials:** The following supporting information can be downloaded at: https://www.mdpi.com/article/10.3390/curroncol31050196/s1, Figure S1: Indications for brain imaging in the initial staging of SCNECC; Figure S2: Practice patterns in the treatment of SCNECC with local relapse (A) and regional nodal relapse (B); Figure S3: Practice patterns in the surveillance of patients with SCNECC after curative intent treatment. File S4: SCNECC Survey; File S5: Individual survey responses from all practitioners.

**Author Contributions:** K.Y.F., H.J.M. and A.S.T. conceptualized the study. K.Y.F., H.J.M. and A.S.T. contributed to software development and implementation. Analyses were performed by K.Y.F., R.C., H.J.M. and A.S.T. Data were collected by K.Y.F., R.C., A.Y.W., A.S., H.J.M. and A.S.T. Curation of data

was conducted by K.Y.F., R.C., A.Y.W., A.S., H.J.M. and A.S.T. The original draft was written by K.Y.F., H.J.M. and A.S.T. Reviewing and editing of the manuscript was performed by K.Y.F., R.C., A.Y.W., A.S., H.J.M. and A.S.T. Data visualization was conducted by K.Y.F. and R.C. Project management and coordination were performed by K.Y.F., H.J.M. and A.S.T. All authors have read and agreed to the published version of the manuscript.

**Funding:** This research received no external funding.

**Institutional Review Board Statement:** The study was conducted in accordance with the Declaration of Helsinki. Ethical review and approval were waived for this study, as this project received a score of 2 (minimal risk) based on ARECCI Ethics Screening tool [9].

**Informed Consent Statement:** Informed consent was obtained from all subjects involved in the study.

**Data Availability Statement:** We will provide our data for independent analysis by a team selected by the Editorial Team for the purposes of additional data analysis or to ensure the reproducibility of this study in other centers, if this is requested.

**Conflicts of Interest:** H.J.M. declares payment or honoraria for lectures, presentations, speakers' bureaus, manuscript writing or educational events with Merk, AstraZeneca, and GlaxoSmithKline; however, these funders had no role in the design of the study; in the collection, analyses, or interpretation of data; in the writing of the manuscript; or in the decision to publish the results. No conflicts of interests are present for other authors.

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
