# Peer review of "Pan-Canadian Analysis of Practice Patterns in Small Cell Carcinoma of the Cervix: Insights from a Multidisciplinary Survey"

_curroncol, doi:10.3390/curroncol31050196_

Round 1
Reviewer 1 Report
Comments and Suggestions for Authors
Thank you for this interesting survey report. A couple of questions:
1. is it possible to know what the response rate was?
2. You have asked individuals for their opinion/practices. Did you ask if persons / groups discussed these cases at tumour board and how did that affect plans.
3. Did your questions evaluate adherence to guidelines such as NCNN or was this only evaluated after data acquisition. i.e. did you ask" in treatment planning do you adhere to guidelines such as NCNN" ?
Author Response
Dear reviewer, thank you for taking the time to review our manuscript and for providing your comments. We have done our best to address these comments; please see below, as well as the revised manuscript. Please let us know if you have further comments and we are happy to provide any additional revisions as needed.
Thank you for this interesting survey report. A couple of questions:
- is it possible to know what the response rate was?
Since this survey was disseminated via email throughout the Society of Gynecologic Oncology of Canada, unfortunately it was not possible to record the total number of clinicians that had access to our survey. We have addressed this as a limitation in the Discussion section (please see revised manuscript, lines 303-306).
- You have asked individuals for their opinion/practices. Did you ask if persons / groups discussed these cases at tumour board and how did that affect plans.
While this would have been a valuable question to include, in our survey we did not ask participants specifically whether they discussed these cases at tumour board. To address this, we have added in our Discussion section a statement to advocate for the need for multidisciplinary discussion at tumour boards (please see revised manuscript, lines 313-316).
- Did your questions evaluate adherence to guidelines such as NCNN or was this only evaluated after data acquisition. i.e. did you ask" in treatment planning do you adhere to guidelines such as NCNN" ?
Our questions did not specifically ask clinicians whether they adhere to current guidelines; rather, we first collected clinicians’ responses and then compared them to existing guidelines afterwards. We have added a statement at the beginning of the Discussion section to clarify our approach (lines 232-234).
Reviewer 2 Report
Comments and Suggestions for Authors This article is about the practice patterns in small cell carcinoma of the cervix in Canada. The section of introduction was well written and explained the situation of SCNECC. Although there was the limitation that the most respondents’ area of expertise was only in one treatment modality, it seems that this could be addressed by adding it to the supplementary file.
1. To be honest, this was the first time I’ve heard about ARECCI Ethics score. It may be a score used in Canada, but since it doesn’t seem to be a standard widely used globally, I think some explanation may be needed.
2. As the authors stated in the discussion, most respondents probably specialize in one of treatment modality. The authors wrote that 'This was addressed by providing the answer choice ''I don't know'' for treatment-related questions. Reassuringly, there was no statistical difference between the practice patterns of the three specialties (p=0.25)'.
(a)However, there was no option for ‘’I don’t know’ on the question paper in the S4 file. Please explain this.
(b)Please clarify what you mean by ‘practice patterns of the three specialties’.
(c) Considering that the number of each expert was small, I don’t think it can be said that there was no problem because there was no statistically significant difference. How about adding the answers to each question by the specialties as a supplementary file?
Author Response
Dear reviewer, thank you for taking the time to review our manuscript and for providing your comments. We have done our best to address these comments; please see below, as well as the revised manuscript. Please let us know if you have further comments and we are happy to provide any additional revisions as needed.
This article is about the practice patterns in small cell carcinoma of the cervix in Canada. The section of introduction was well written and explained the situation of SCNECC. Although there was the limitation that the most respondents’ area of expertise was only in one treatment modality, it seems that this could be addressed by adding it to the supplementary file.
- To be honest, this was the first time I’ve heard about ARECCI Ethics score. It may be a score used in Canada, but since it doesn’t seem to be a standard widely used globally, I think some explanation may be needed.
Thank you for bringing this to our attention. The ARECCI Ethics score is a 34-question-based risk score developed, validated and employed in Canada, that determines the level of risk, types of ethical risks, and the appropriate type of ethics review for research and quality improvement projects. Our project received a score of 2 which is the lowest possible risk category (‘minimal risk’). We have added further clarification in the Methods section of the revised manuscript (lines 80-82). If you have a suggestion for another tool which is used in other countries, we would be happy to perform and report a second ethical risk assessment for our project as well; please let us know.
- As the authors stated in the discussion, most respondents probably specialize in one of treatment modality. The authors wrote that 'This was addressed by providing the answer choice ''I don't know'' for treatment-related questions. Reassuringly, there was no statistical difference between the practice patterns of the three specialties (p=0.25)'.
(a)However, there was no option for ‘’I don’t know’ on the question paper in the S4 file. Please explain this.
There is an option for ‘I don’t know’ in Q20 and Q21. As these questions pertain to the choice of palliative systemic therapy, we considered these two questions to be the most esoteric to certain providers who would otherwise primarily prescribe curative intent and locoregional treatment. We have amended our manuscript to clarify that we included the option ‘I don’t know’ for only selected questions, rather than for every question (Please see Discussion section in revised manuscript file, lines 296-299).
(b)Please clarify what you mean by ‘practice patterns of the three specialties’.
By ‘practice patterns of the three specialties’ we meant first line treatment approach for each stage of SCNECC (i.e. early-stage, locally advanced, distantly recurrent and metastatic SCNECC; this corresponds to Q12, Q13, Q20, Q21) selected by each of the three specialties (gynecological oncology, radiation oncology, medical oncology), compared using Chi-Square test, using alpha of 0.05. However, in response to your suggestion in (c) we agree to eliminate this statement and we will instead add the individual answers by each specialty in a supplementary file.
(c) Considering that the number of each expert was small, I don’t think it can be said that there was no problem because there was no statistically significant difference. How about adding the answers to each question by the specialties as a supplementary file?
Thank you for this suggestion. We have removed the statement “Reassuringly, there was no statistical difference between the practice patterns of the three specialties (p=0.25).” In place of this, we have included supplementary file S5 to show the responses to each question by specialty, and we hope that this will be more informative. Please see Discussion section of revised manuscript which has been revised accordingly (lines 298-299).